

# Integrated economic and environmental analysis of agricultural straw reuse in edible fungi industry

Wencong Lu[1], Shuao Yu[1], Yongxi Ma[2] and Hairong Huang[2]

[1] China Academy for Rural Development, Zhejiang University, Hangzhou, Zhejiang, China
[2] School of Economics and Management, Zhejiang Sci-Tech University, Hangzhou, Zhejiang, China

Corresponding author
Yongxi Ma, myx@zstu.edu.cn

## ABSTRACT

**Background:** China currently faces severe environmental pollution caused by burning agricultural straw; thus, resource utilization of these straws has become an urgent policy and practical objective for the Chinese government.

**Methods:** This study develops a bio-economic model, namely, "straw resource utilization for fungi in China (SRUFIC)," on the basis of a field survey of an edible fungi plant in Zhejiang, China, to investigate an integrated economic and environmental performance of straw reuse in fungi production. Five scenarios, which cover changes in the production scale, wage level, and price fluctuations of the main product and inputs, are simulated.

**Results:** Results reveal that (1) the pilot plant potentially provides enhanced economic benefits and disposes added agricultural residues by adjusting its production strategy; (2) the economic performance is most sensitive to fungi price fluctuations, whereas the environmental performance is more sensitive to production scale and price of fungi than other factors; (3) expanding the production scale can be the most efficient means of improving the performance of a plant economically and environmentally.

**Discussion:** Overall, agricultural straw reuse in the edible fungi industry can not only reduce the environmental risk derived from burning abandoned straws but also introduce economic benefits. Thus, the straw reuse in the fungi industry should be practiced in China, and specific economic incentive policies, such as price support or subsidies, must be implemented to promote the utilization of agricultural straws in the fungi industry.

# INTRODUCTION

China is one of the major agricultural nations in the world; this country produces 900 million tons of agricultural straw annually, 20% of global straw output. In rural China, agricultural straws have been used traditionally as an important energy source for cooking and heating. However, farmers currently have access to additional convenient substitute household energy sources, such as electricity and gas. The residue straw is

difficult to dispose of on the field. In 2015, more than 180 million tons of straw were burned or abandoned (*MOA, 2016*). However, burning straw releases inhalable particulate matter (PM10 and PM2.5), greenhouse gases ($CH_4$ and $CO_2$), nitrogen oxides (NOx), and sulfur dioxide ($SO_2$), which brings heavy air pollution problems and severely affected ecological systems and human health in China (*Shi et al., 2014*). Moreover, agricultural straws can be used as biofuel, fodder, and fertilizer. Therefore, burning straws not only damages the environment but also wastes valuable resources.

The huge amount of wasted resources and the severe pollution of the environment caused by burning agricultural straw warn the Chinese government of the urgent requirement for developing new strategies for disposing of and reusing agricultural straws. In 1999, the Ministry of Environmental Protection (MEP) of China announced the "Regulations on straw burning prohibition and comprehensive utilization," prohibits straw burning in certain areas and proposes a target ratio (60%) of comprehensive utilization (*MEP, 1999*). Then, the "Renewable Energy Law" and the "Circular Economy Promotion Law," encourage the comprehensive utilization of agricultural straws (*NPC, 2005*, *2008*). The National Development and Reform Commission (NDRC) of China formulated a "Medium and Long-term Development Plan for Renewable Energy in China" (*NDRC, 2007*), and then China's State Council (CSC) issued the "Notice on Promoting Comprehensive Utilization of Crop Straws" to clearly demonstrate the straw program for each authority level (*CSC, 2008*). The number of straw burning sites has effectively decreased given the implementation of these policies; however, the immense challenge of utilizing massive amounts of straws still. Then, the "Circular Economy Development Strategy" (*CSC, 2013*), "Technology Catalog of Crop Straw Comprehensive Utilization" (*NDRC & MOA, 2014*), and "Instruction on Further Promoting Comprehensive Utilization of Crop Straw" (*NDRC, 2015*) were introduced to update and supplement the abovementioned policies; these policies also highlighted five major categories of utilization, that is, fertilizers for crops, feeds for livestock, feed-stocks for industry, media for fungi, and biomasses for energy. Furthermore, these policies have indicated that agricultural straws in China shall be "reused as recycled resources (RRR)," which can not only reduce negative environmental impacts but also increase economic benefits. The resource utilization of agricultural straws has become a significant policy and practical objective for the Chinese government (*MOA, 2017*).

Many previous studies have discussed the techniques involved in the five categories of straw utilization; numerous studies focus on energy utilization in terms of environmental or economic effects. Research in developed and developing countries have verified that applying straw-derived electricity, gas, and bioethanol can reduce the global warming potential to the world and generate much less human- or eco-toxicities than traditional fossil energies, such as coal-fired electricity and natural gas (*Nguyen, Hermansen & Mogensen, 2013*; *Nguyen, Hermansen & Nielsen, 2013*; *Soam et al., 2016*; *Song, Song & Zhang, 2016*; *Van Nguyen et al., 2016*; *Weiser et al., 2014*). Moreover, the energy utilization of agricultural straws is economically viable and efficient in East Europe (*Cosic, Stanic & Duic, 2011*; *Dodic et al., 2012*; *Zbytek et al., 2016*) and Asia (*Delivand et al., 2012*; *Singh, 2016*; *Sun et al., 2017*), especially in terms of the price of biomass products and

costs incurred in collection, processing, and facilities. For the utilization of fertilizer, several long-term field experiments have confirmed that straw compost returning can improve the nutrient cycling structure of farming land using an intensive cropping system and reduce the risk of excessively fertilized soils (*Kim, Choo & Cho, 2017*; *Roca-Perez et al., 2009*; *Tian et al., 2016*; *Zheng et al., 2015*). Another branch of studies investigates greenhouse gas (GHG) emissions from straw returning, calculates global warming potentials of different returning modes (*Hu et al., 2016*; *Wu et al., 2015*; *Zhang et al., 2015*). *Monteleone et al. (2015)* examine an optimal trade-off between "straw to soil" and "straw to energy," the results show that the use of straw for energy generation consistent with optimizing the cropping system are key factors for long-term environmental sustainability in terms of the GHG reduction and fossil displacement. However, very limited studies have accomplished an environmental or economic analysis related to the other three categories of straw utilization. Feeding animals with pretreated straws (e.g., silage and ammonification) can possibly reduce the GHG emission from livestock (*Fan, Yang & Li, 2006*). *Xi & Zhou (2015)* introduce a circular economy of agricultural straw utilization as substrates for edible fungi, but no quantitative analysis of environmental or economic effects has been conducted.

Most studies have adopted a life cycle approach to assess the environmental and economic performances of straw utilization (*Clare et al., 2015*; *Delivand et al., 2012*; *Hong et al., 2016*; *Kunimitsu & Ueda, 2013*; *Nguyen, Hermansen & Mogensen, 2013*; *Nguyen, Hermansen & Nielsen, 2013*; *Song et al., 2017*). This approach can clearly indicate the input and output at each phase of the life cycle of the objective product or process of utilization. However, the life cycle approach is essentially a descriptive method that relies on life cycle inventory data and does not reveal the dynamic, interactive relationship between economic behavior and environmental systems. A possible alternative method is applying bio-economic (or environmental-, ecological-economic) models, which have been widely accepted by agricultural or ecological economists when analyzing the integrated effects of environmental and economic systems (*Arfini, 2012*). A bio-economic model, which merges biophysics and economics, can investigate the multi-disciplinary and multi-scale effects of a given problem through biophysical equations and economic programming methods (*Filchman, Louhichi & Boisson, 2011*).

This paper develops a specific model based on a bio-economic framework to combine the environmental and economic aspects into an integrated analysis of agricultural straw utilization in the study of a fungi enterprise from Zhejiang, China. This study aims to optimize the quantity of disposed agricultural straws (environmental effect) and maximize the economic returns (economic effect) simultaneously under subjective constraints. The economic and environmental effects are assessed by bio-economic model simulations under different scenarios. Sensitivity analysis is also incorporated into the model simulations to derive the potential impact from changes in the production scale and price of input materials and products.

The rest of this paper is organized as follows: Section "Study Case" introduces a profile of the participating fungi enterprise and the process of fungi production; Section "Integrated Economic and Environmental Analysis" describes the bio-economic model

for integrated environmental and economic analysis for agricultural straw utilization in the fungi industry, along with 12 scenarios of model simulation; Section "Results and Discussion" presents the simulation results of the environmental and economic effects in different scenarios and discusses several critical points with respect to practical recommendations. Section "Conclusion and Policy Implications" summarizes the results of the study and highlights several policy implications.

## STUDY CASE

This study was conducted in a pilot edible fungi plant in Jinhua City, Zhejiang Province, China. The pilot plant was established in 2004, and currently, the pilot plant has 5,000 m$^2$ of fungi growing room with two lines of production. The pilot plant has disposed of 1,080 tons of straw and 2,520 tons of other residues (including cottonseed hulls, bagasse, bran, and cornstarch) in 2015. A total of six million bags of fungi and 3,600 tons of edible fungi are produced, thereby generating a considerable economic return of CHY[1] 22.2 million, along with CHY 43.4 million total benefit and CHY 21.2 million total cost. The pilot plant also provides 85 fixed work positions and dozens of flexible part-time jobs to the local community. The residue straw (270 ton) was burned in the field that brought heavy air pollution.

In the past decade, the pilot plant has developed an integrated, environmentally friendly processing system to produce fungi (Fig. 1). All masses of fungi growth medium are agricultural or forestry wastes; 80% of which are crop straws and cottonseed hulls; the remaining 20% consists of bagasse, bran, and cornstarch. The first step of growing fungi is preparing nutrient growth medium by proportionately mixing the compost of crop straw, cottonseed hulls, and bagasse proportionately. The composting procedure in the medium preparation phase lasts six months. The fertile fungi medium is packed into fungi bags. Each fungi bag must be autoclaved and sterilized in a furnace for at least 4 h. Then, the fungi spawns are placed into the cavities for mycelium incubation. This phase lasts for approximately 100–120 days. Finally, the fungi are harvested.

The used medium and remaining fungi residue are generally utilized in four ways. The residues are (1) collected as fertile materials for secondary fungi bags (20–30%), (2) delivered to livestock or poultry farms as organic feeds (30–40%), (3) mixed with the sewage generated during fungi growing as fertilizers (20–30%), and (4) dried out into biofuels (10%–20%).

## INTEGRATED ECONOMIC AND ENVIRONMENTAL ANALYSIS

### Modeling

Bio-economic models are widely used on an individual or multiple agricultural production systems of the crop, livestock, and fisheries because these models can be applied across borders of disciplines between biophysical equations and economic behaviors (*Arfini, 2012*; *Janssen & Van Ittersum, 2007*; *Kragt et al., 2016*). Linear programming method is commonly selected as the mathematical approach to solving optimization problems in bio-economic models for the following reasons: this method

[1] CHY: Chinese Yuan, China's currency unit.

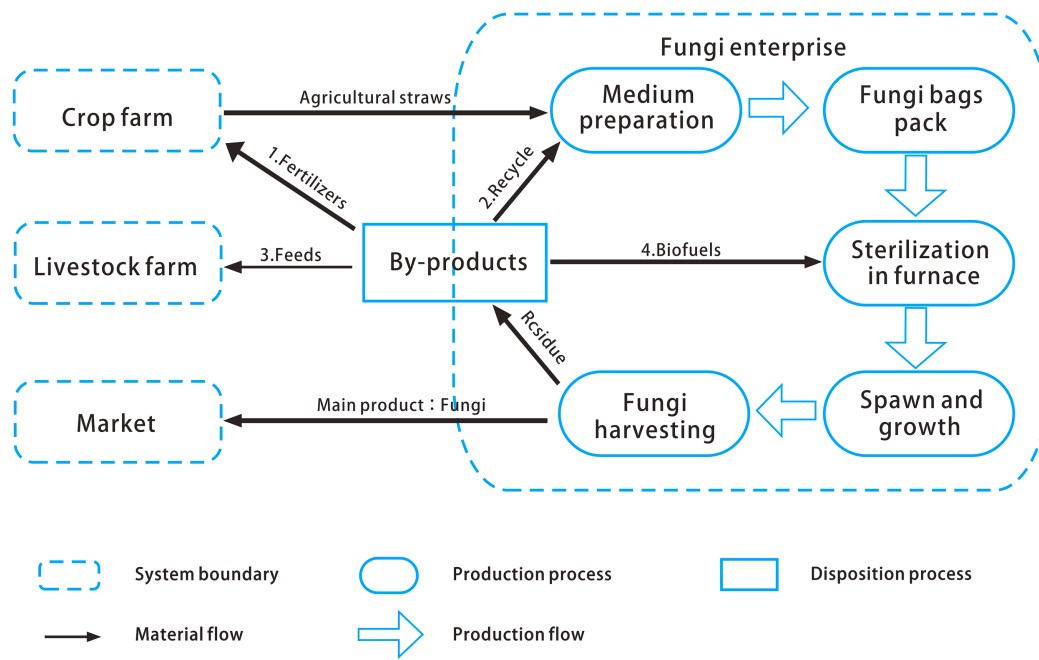

**Figure 1 Edible fungi production system of the pilot plant.** The first step is preparing nutrient growth medium by proportionately mixing the compost of crop straw, cottonseed hulls, and bagasse proportionately. The composting procedure in the medium preparation phase lasts six months. Then, the fertile fungi medium is packed into fungi bags. Each fungi bag must be autoclaved and sterilized in a furnace for at least 4 h. Thereafter, the fungi spawns are placed into the cavities for mycelium incubation. This phase lasts for approximately 100–120 days. Finally, the fungi are harvested.

enables multi-restrictions to be considered simultaneously, offers a specific and valid goal-seeking procedure, and outputs model results easily when variables are changing (*Lu, Ma & Bergmann, 2014*; *Ma, Lu & Bergmann, 2014*). For this study, the "straw resource utilization for fungi in China (SRUFIC)" model based on the general structure of the "ecological-economic" model (*Wossink, Oude Lansink & Struik, 2001*; *Pacini et al., 2004*) is constructed to analyze the integrated environmental and economic effects in the pilot enterprise.

$$\pi(X, Y, Z, B; \theta) = \text{MAX}\{[\text{Pr} \times Y - W \times X] - C\} \tag{1}$$

$$\text{S.T.} : \text{T}(X) = \{Y | (Y, X; \theta) \in T\} \tag{2}$$

$$\text{G}(Z, Y, X; \theta) \geq G_0 \tag{3}$$

$$B \leq \bar{B} \tag{4}$$

Equation (1) expresses the profit maximization function of the SRUFIC model in terms of vectors of material input ($X$), product output ($Y$), environmental indicator ($Z$), resource budget ($B$), technological level ($\Theta$), fixed cost ($C$), price of products (Pr), and price of inputs ($W$). The fixed cost, technological level, and prices are considered constants in one stage of the programming. Equations (2)–(4) are model subject constraints to the

programming. The fungi production function, T($X$), is defined as a transformation process that combines materials and a group of products under a certain technological level. G($Z$, $Y$, $X$; Θ) refers to the ultimate environmental effects of the fungi production system with respect to the given indicators of input, output, environment, and technology. Thus, the "zero emission" target can be realized in the production system because all production residues, including sewage, can be utilized in several ways. The ultimate environmental effect is described as the "treatment effect on agricultural residues (straws) in fungi production," or the quantity disposed of by the plant in general. $G_0$ is the benchmark of this effect. Material balance approach (MBA) is used to build a biophysical formulation that traces the environmental effects of the fungi production system. The MBA means that materials in a physical system are not lost, and that material inputs in processes end up in either stock accumulation or material output flows (*Nijkamp & van den Bergh, 1997*). All input nutrients result in final products or wastes (*Ma, Lu & Bergmann, 2014*). The most important nutrients, that is, nitrogen (N) and phosphorus (P), are considered in the MBA because excess N and P are also major pollution elements for soil and water.

$$R = \sum_i^n x_i(N, P, K) - \sum_j^m y_j(N, P, K), \tag{5}$$

where $R$ represents the portion released into the surroundings. $R$ can be considered less than or equal to zero because all the residues and wastes are utilized as by-products in this study.

In the SRUFIC model, $B \leq \bar{B}$ denotes that finite producing resources are available in the production system. Credit and land are important factors which increases the production and income of the farmers (*Hussain, 2012*). In rural areas of China, small farms face many difficulties getting enough loan credit and land to expand their production capacity. The main resource constraints considered in this study include liquid capital and production capacity.

Liquid capital constraint. The fungi enterprise should have a full payment capability for expenditures on the materials of a medium, spawns, salary of workers, electricity, and freshwater to sustain the production system.

$$\sum_i^n e_i \leq \bar{E}, \tag{6}$$

where $e_i = w_i \times x_i$; $e_i$, $w_i$, $x_i$ are the expenditure, price, and quantity of input I, respectively; $n$ is the number of inputs; and $\bar{E}$ is the maximum amount of viable liquid capital.

Production capacity constraint. Facilities in the enterprise undergo regular maintenance; thus, no additional production line can be equipped.

$$\sum_j^m y_j \leq \bar{Y}, \tag{7}$$

where $y_j$ refers to the output of product $j$, $m$ is the number of products (including by-products), and $\bar{Y}$ is the current maximum production capacity.

## Data source

The primary data were obtained through a field survey of the pilot plant in 2016. The frontline workers, technical staff, department managers, and enterprise owners participated in the survey. A detailed production dataset that covers economic and environmental performances has been established based on several annual and seasonal reports from 2015. The inputs (including crop straw, cottonseed hulls, and bagasse), products, and by-products were calculated by dividing the value by quantity. The nutrient content data and technical parameters used in this study were offered by the technical staff from the plant.

## Simulation scenarios

Five simulation scenarios are constructed in this study to evaluate the environmental and economic effects of straw utilization in the fungi production under different market situations. Each scenario covers the two-way fluctuation of the study variables.

Scenario I: Fungi price. The fungi price is assumed to fluctuate in this scenario. Fungi price is highly related to the total economic income of this pilot plant and reflects market demand and supply information, which can impact the production decisions of enterprise owners.

Scenario II: Agricultural straw and Scenario III: Cottonseed hull. The two scenarios consider the price fluctuation of the two main material inputs, namely, agricultural straw and cottonseed hull. These inputs represent 80% of the weight of the fungi growth medium. The prices of straw and hull comprise the largest portion of the total production cost. The quantities of straw and hull that the fungi plant disposes of are also determined by these prices.

Scenario IV: Wage. This scenario considers the changes in labor price. A high (or low) wage level may affect the employment decisions of enterprise owners and increase (or decrease) the expenditure on salaries.

Scenario V: Scale. In this scenario, the production capacity constraint in the SRUFIC model is loosened. The pilot plant is allowed to add new production facilities to extend the input scale or reduce workload to reduce the input scale. The potentially technical parameters are assumed to be unchanged in this scenario (*Smith, Card & Young, 2006*).

# RESULTS AND DISCUSSION

## Simulation results

### Output and input

A general algebraic modeling system is used to solve the programming results of the SRUFIC model. Model calibration and validation were conducted previously to check the consistency between real situations and model scenario solutions. The scenarios studied are possible future market conditions and possible production scales. "Status quo" represents the actual situation according to the pilot plant survey. "Base" scenario refers to the optimized results calculated by a model mathematical programming, which represents the optimal production situations under present constraints. Moreover, Optimization

**Table 1 The baseline data in the case study.**

|  | Price | Quantity |
| --- | --- | --- |
| **Output** | | |
| Fungi | 12 (CHY/kg) | 3,882 (ton) |
| By-products | 0.1 (CHY/kg) | 1,519 (ton) |
| **Input** | | |
| Straw | 0.601 (CHY/kg) | 1,146.23 (ton) |
| Seed hull | 1.8 (CHY/kg) | 1,811.78 (ton) |
| Bagasse | 0.3 (CHY/kg) | 406.73 (ton) |
| Bran | 1.3 (CHY/kg) | 184.88 (ton) |
| Cornstarch | 2.8 (CHY/kg) | 147.9 (ton) |
| Spawn (CHY/kg) | 1 (CHY/kg) | 1,450 (ton) |
| Employment | 55,200 (CHY/year) | 85 |
| Electricity | 0.57 (CHY/kvh) | $4.66 \times 10^6$ (kvh) |
| Fresh water | 2.2 (CHY/ton) | $2.15 \times 10^4$ (ton) |

Note:
The prices and quantities of output and input which described the production function in the pilot plant.

results are considered the baseline for succeeding simulation scenarios (*Smith, Card & Young, 2006*). The prices and quantities of output and input which described the production function in the plant are reported in Table 1.

Optimization can increase the harvest by 282 tons of fungi and 115 tons of by-products more than "Status quo" after adding 98 tons of medium and 10 tons of spawn. This result reveals that the pilot plant is capable of expanding its production intensity under the current set of constraints. Meanwhile, there is a better environmental effect by no straw burning.

Table 2 presents the results of fungi production output and input under different simulation scenarios.

For the fungi price scenario, a high market price urges the plant to improve its production intensity. If the price of fungi increases by 7%, then the pilot plant will hire six more workers than base scenario. On the contrary, the input and output will decline if the price of fungi decreases.

In Scenarios II and III, the input price decrease allowed the plant to purchase additional production materials. The growth extents, especially in Scenario III, although both rows of "price decrease 10%" in agricultural straw and cottonseed hull have shown rises in each column. The straws are evidently smaller than in previous scenarios. One possible explanation might be due to agricultural straws and cottonseed hulls are generally less expensive, and their prices are not as elastic. Thus, plant owners are less sensitive to straws and hulls than to the other elements of production decisions. In Scenario IV, low wage levels will stimulate the pilot plant to use additional workers as presumed, thereby leading to an upward trend of initial inputs and final outputs. From the perspective of production, the changes in output and input caused by the decline and upraise in wages are relatively close; both of which are smaller than in Scenario III.

**Table 2 Simulation results of output and input quantities.**

| Scenario | Output | | Input | | | | |
|---|---|---|---|---|---|---|---|
| | Fungi (ton) | By-pro. (ton) | Medium (ton) | Spawn (ton) | Employ. | Electricity (kvh) | Water (ton) |
| **Base (Optimization)** | 3,882 | 1,519 | 3,698 | 1,450 | 85 | $4.66 \times 10^6$ | $2.35 \times 10^4$ |
| **Scenario I: Fungi** | | | | | | | |
| *Price decrease 10%* | 3,652 | 1,429 | 3,478 | 1,364 | 80 | $4.38 \times 10^6$ | $2.21 \times 10^4$ |
| *Price increase 10%* | 4,156 | 1,626 | 3,959 | 1,552 | 91 | $4.99 \times 10^6$ | $2.51 \times 10^4$ |
| **Scenario II: Ag. Straw** | | | | | | | |
| *Price decrease 10%* | 3,914 | 1,531 | 3,727 | 1,462 | 86 | $4.70 \times 10^6$ | $2.37 \times 10^4$ |
| *Price increase 10%* | 3,866 | 1,513 | 3,682 | 1,444 | 85 | $4.64 \times 10^6$ | $2.34 \times 10^4$ |
| **Scenario III: Ct. seed hull** | | | | | | | |
| *Price decrease 10%* | 4,005 | 1,567 | 3,811 | 1,496 | 88 | $4.81 \times 10^6$ | $2.42 \times 10^4$ |
| *Price increase 10%* | 3,786 | 1,481 | 3,575 | 1,414 | 83 | $4.54 \times 10^6$ | $2.29 \times 10^4$ |
| **Scenario IV: Wage** | | | | | | | |
| *Decline 10%* | 4,062 | 1,589 | 3,869 | 1,517 | 89 | $4.87 \times 10^6$ | $2.47 \times 10^4$ |
| *Upraise 10%* | 3,739 | 1,463 | 3,561 | 1,397 | 82 | $4.49 \times 10^6$ | $2.26 \times 10^4$ |
| **Scenario V: Scale** | | | | | | | |
| *Shrink 10%* | 3,402 | 1,331 | 3,240 | 1,271 | 75 | $4.08 \times 10^6$ | $2.06 \times 10^4$ |
| *Expand 10%* | 4,158 | 1,627 | 3,960 | 1,553 | 92 | $4.99 \times 10^6$ | $2.51 \times 10^4$ |

**Note:**
The results of fungi production output and input under five simulation scenarios, including four price scenarios (fungi, straw, seed hull, and labors' wage) and one scale scenario. Higher market (output) prices and lower input prices can urge the plant to improve its production intensity. Besides, the production behaviors of the pilot plant are more sensitive to scale shrink than to scale expansion; moreover, the pilot plant is in the phase of decreasing returns to scale.

In Scenario V, a scale shrink of 10% results in the decrease in every column, and the reverse occurs in terms of scale expansion. The impact is more obvious in scale shrink than in scale expansion. For example, fungi production increases by 276 tons (less than 10%) in "expand 10%," but decreases by 480 tons (more than 10%) in "shrink 10%." These results reveal that the production behaviors of the pilot plant are more sensitive to scale shrink than to scale expansion; moreover, the pilot plant is in the phase of decreasing returns to scale.

### *Economic and environmental performances*

Table 2 presents the results of the economic and environmental performances in the pilot fungi plant under different simulation scenarios. In this paper, economic performance refers to the total benefit, cost, and net benefit derived from the fungi plant. Environmental performances refer to the quantities of agricultural straw and other agricultural residues disposed of, such as cottonseed hull and bagasse. Optimization enables the plant to generate more total benefit growth (CHY 3.3 million) than total cost growth (CHY 0.1 million) than Status quo, thereby generating an increase of CHY 3.2 million net benefit. Furthermore, instead of burning 270 tons of straws in status quo, the plant under Optimization, which assumes "zero straw burning," will dispose additional 66 tons of straw, and the rest (204 tons) is preferably returned back to the field. The pilot plant shall adjust its production strategy to obtain additional profit and simultaneously dispose of additional agricultural straw for the local community.

**Table 3 Simulation results of economic and environmental performances.**

| Scenario | Economic (million CHY) | | | Environmental (ton) | |
|---|---|---|---|---|---|
| | Total benefit | Net benefit | Cost | Straw disp. in plant | Straw to field |
| **Base** | 46.7 | 25.4 | 21.3 | 1,146 | 204 |
| **Scenario I: Fungi** | | | | | |
| *Price decrease 10%* | 39.6 | 19.3 | 20.3 | 1,078 | 272 |
| *Price increase 10%* | 55.0 | 32.6 | 22.4 | 1,227 | 123 |
| **Scenario II: Ag. Straw** | | | | | |
| *Price decrease 10%* | 47.1 | 25.7 | 21.4 | 1,156 | 195 |
| *Price increase 10%* | 46.5 | 25.2 | 21.3 | 1,141 | 209 |
| **Scenario III: Ct. seed hull** | | | | | |
| *Price decrease 10%* | 48.2 | 26.7 | 21.5 | 1,182 | 168 |
| *Price increase 10%* | 45.6 | 24.4 | 21.2 | 1,118 | 232 |
| **Scenario IV: Wage** | | | | | |
| *Decline 10%* | 48.9 | 27.3 | 21.6 | 1,199 | 151 |
| *Upraise 10%* | 45.0 | 23.8 | 21.2 | 1,104 | 246 |
| **Scenario VI: Scale** | | | | | |
| *Shrink 10%* | 41.0 | 21.7 | 19.3 | 1,004 | 346 |
| *Expand 10%* | 50.1 | 27.6 | 22.5 | 1,228 | 122 |

**Note:**

The plant is motivated to increase input and expand its output to generate high income when the output price increases. Among the three input price scenarios, wage changes can cause the most significant fluctuations in each economic or environmental column. Straw price changes exhibit the least impact on both performances, and the cottonseed hull scenario lies in the middle.

Similar to Table 1, the Optimization results in Table 3 are then considered a reference for succeeding simulation scenarios.

Scenarios I–IV are essentially price scenarios, with Scenario I relating to output price and the other three relating to input price. The plant is motivated to increase input and expand its output to generate high income when the output price increases. In Scenario I, a 10% fungi price increase exhibited significant positive effects on the economic and environmental performances. The net income grows by 28% to CHY 3.6 million, and the sum amount of disposition increases by 237 tons, including 81 tons of agricultural straws. However, if input prices increase, then the plant shall have less payment capability to hire as many workers as before or purchase the same amount of materials. Among the three input price scenarios, wage changes can cause the most significant fluctuations in each economic or environmental column. Straw price changes exhibit the least impact on both performances, and the cottonseed hull scenario lies in the middle.

A large production scale can induce increases in the total benefit, cost, and net benefit and dispose of additional 82, 128, and 29 tons of agricultural straw, cottonseed hull, and bagasse residue, respectively. The economic performance is similar to the output and input results discussed previously because economic items are equal to quantities multiplied by constant prices. This result may reveal that the fungi production behavior of the pilot plant under the existing conditions fits the law of diminishing marginal returns.

**Table 4 Sensitivity results of economic and environmental performances.**

|  | Fungi | | Ag. straw | | Ct. seed hull | | Wage | | Scale | |
| --- | --- | --- | --- | --- | --- | --- | --- | --- | --- | --- |
|  | Down | Up | Down | Up | Down | Up | Down | Up | Down | Up |
| Total benefit | −1.60 | 1.67 | −0.01 | −0.13 | 0.23 | −0.33 | 0.37 | −0.45 | −1.31 | 0.62 |
| Net benefit | −2.43 | 2.76 | 0.12 | −0.08 | 0.49 | −0.42 | 0.73 | −0.61 | −1.47 | 0.82 |
| Cost | −0.45 | 0.54 | 0.04 | 0.01 | 0.09 | −0.04 | 0.13 | −0.07 | −0.93 | 0.57 |
| Employment | −0.59 | 0.71 | 0.12 | −0.12 | 0.35 | −0.24 | 0.47 | −0.35 | −1.18 | 0.82 |
| Straw disposal | −0.59 | 0.71 | 0.08 | −0.04 | 0.31 | −0.25 | 0.46 | −0.37 | −1.24 | 0.71 |

Note:
Expanding the production scale can be an efficient means of achieving high economic returns and dispose of additional agricultural straws because market prices and average wage levels cannot be controlled by a single plant.

## Sensitivity analysis

Comparisons between Status quo and Optimization reveal that the pilot fungi plant can obtain more output under its current budget and other constraints. The plant shall adjust its producing strategy by generating extra medium and inputting additional spawn in the production system to improve economic benefit and dispose of large quantities of agricultural residues for the local community. For the rest of the scenarios, sensitivity results in Table 4 provide a clear picture of the economic and environmental performances of the pilot plant. The sensitivity results are exactly the same among the inputs because the medium proportion ratios of straw, hull, and bagasse are assumed to be constant. "Ag. Straw" is listed to represent environmental performances.

In terms of economic performance, the total benefit and net benefit is most sensitive to fungi price fluctuations, followed by production scale and wage changes. Product price is key factor for the better economic performance. But for the cost and employment, the scale has the largest impact, followed with fungi price, wage and the other agricultural residues prices. For the environmental performance, the disposal of agricultural straws is more varied in scale and fungi scenarios than in other columns. The agriculture price has a minimal effect on the economic and environmental performance. These results may reveal that expanding the production scale can be an efficient means of achieving high economic returns and dispose of additional agricultural straws because market prices and average wage levels cannot be controlled by a single plant.

## CONCLUSIONS AND POLICY IMPLICATIONS

The reuse of agricultural straws in the fungi industry is crucial in terms of reducing environmental pollution and promoting efficient resource utilization in agricultural production. A holistic, integrated bio-economic model is used in this study to achieve a comprehensive economic and environmental analysis. The fungi production and agricultural straw disposal systems are incorporated. The case study indicates that the resource utilization of agricultural straws in the edible fungi industry is feasible for economy and environment. The reuse of agricultural straws in the fungi industry can minimize the environmental risks of burned or abandoned straw and avoid "secondary pollution" in the process of straw treatment.

This paper presents the following conclusion on the basis of the simulation analysis in the case study: the current use of straw resources is inefficient in the pilot fungi plant. The economic and environmental performances can be improved by increasing the medium and spawn inputs under the current production conditions. A large production scale increases income and improves environmental performance. However, the production behaviors of the pilot plant are more sensitive to scale shrinking than to scale expansion. An increase in the output price will stimulate the plant production and dispose of added agriculture straws. However, the economic and environmental performances of fungi production will deteriorate if the input prices increase. The sensitivity analysis also indicates that the economic performance is most sensitive to fungi price fluctuations and that the environmental performance is more sensitive to production scale and price of fungi than other factors.

From the perspective of policy instruments, devoting extra effort to the comprehensive resource utilization of agricultural straws is necessary for Chinese farms to solve the problems of environmental pollution caused by leftover straw. Technological and environmental economic means should be applied comprehensively to "convert wastes into resources," promote economic and environmental benefits, and achieve sustainable development for agriculture. The reuse of agricultural straws in the fungi industry should be demonstrated and extended to suitable areas in China. The expansion of the production scale in the fungi plants should be encouraged to enhance economic profits and improve environmental performances. Moreover, specific economic incentive policies, such as price support and subsidies, must be implemented to promote the resource utilization of agricultural straws in the fungi industry.

### Funding
This study was funded by The National Social Science Fund of China (No. 14BGL206) and the Project of Philosophy and Social Sciences in Zhejiang Province (No. 15LLXC23YB). The funders had no role in study design, data collection and analysis, decision to publish, or preparation of the manuscript.

### Grant Disclosures
The following grant information was disclosed by the authors:
The National Social Science Fund of China: 14BGL206.
Project of Philosophy and Social Sciences in Zhejiang Province: 15LLXC23YB.

### Competing Interests
The authors declare that they have no competing interests.

### Author Contributions
- Wencong Lu conceived and designed the experiments, contributed reagents/materials/ analysis tools, authored or reviewed drafts of the paper, approved the final draft.

- Shuao Yu performed the experiments, analyzed the data, contributed reagents/materials/analysis tools, prepared figures and/or tables, authored or reviewed drafts of the paper, approved the final draft.
- Yongxi Ma performed the experiments, analyzed the data, contributed reagents/materials/analysis tools, prepared figures and/or tables, authored or reviewed drafts of the paper, approved the final draft.
- Hairong Huang analyzed the data, contributed reagents/materials/analysis tools, approved the final draft.

## Data Availability

The raw data are included in Tables 1 and 2.

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
