# Peer review of "Integrated economic and environmental analysis of agricultural straw reuse in edible fungi industry"

_PeerJ, doi:10.7717/peerj.4624_

## Round 0.1 · original submission · Minor Revisions

Please revise the manuscript and carefully respond to the comments of both reviewers.

Reviewer 1 ·

Basic reporting

The manuscript is clearly written in professional language, and the structure, figures and tables are professional. However, many grammar issues still exist, the English language should be improved by professional institute if necessary.
In Line 45, CH4, CO2, Nox, SO2, such chemical formulas should be standardized with subscript like CH4.

Experimental design

Line 99-118, this paragrapgh contains too many words, the aim of this study should be emphasized with a single paragraph.

There is a weakness in the statistical analysis, a lack of descriptive introduction of the sample, which is useful to future readers. It should be improved upon before Acceptance.

Line 144-147, the author said there are four ways…, but in Figure 1, the fourth path (dried out into biofuels) was not displayed, which should be improved upon before acceptance.

Validity of the findings

I commend the author for his/her extensive and excellent work, they compiled the survey data and annual/seasonal reports of the enterprise, the data is robust, statistically sound.

line 323, "......is feasible for technology, economy and environment". I can not find any related information about "feasible for technology" in the text, please supplement explanation for it, or remove the word "technology,".

Additional comments

The authors seem to have done much work on disposal of or reuse of agricultural straw, it has environmental and economic benefits. And also a bio-economic model –“straw resource utilizaiton for fungi in China” was introduced, which is described with sufficient detail and information. So this work is interesting and is of important significance. However, some typographic errors and discrepancies are found needed to correct in the paper. In addition, some part of the sentences need more clarification for easy understanding therefore the following are some of the corrections recommended for enhancement.
For more information, pls refer to the original manuscript PDF document.

Annotated reviews are not available for download in order to protect the identity of reviewers who chose to remain anonymous.

Reviewer 2 ·

Basic reporting

The article is overall well written, though the writing can be improved. There are places throughout the paper with some grammar mistakes.

Experimental design

The method is overall adequate. However, I think more clarifications are needed for the optimization model. For instance, what type of production function did you use in the model? What are the baseline model parameters?

Validity of the findings

The results make sense. I have a few comments regarding the interpretation of the results. See my general comments to the authors.

Additional comments

1. It would be helpful to provide a table of the baseline data used in the optimization model. For instance, what are the prices assumed in the model? What is the production function?

2. What is the “Material balance approach”?

3. One thing the authors want to consider is to include the cost of capital in the profit function. Based on my reading, the objective function in the optimization model does not include the cost of capital (opportunity cost of liquid capital). Given the scale of the production considered in your analysis, the capital cost may turn out to be rather large.

4. In scenario 1, when you relax the production capacity constraint, do you also revise your capital constraint, and potentially other parameters used in the model?

5. Table 1 scenario 3, why does the water usage go up when the price of fungi goes down by 10%?

6. Based on table 2, fungi price fluctuations appear to have the largest impact on the net profit, followed by scale, wage, and others. However, in table 3 however, you show that the production scale has the largest effect on the net benefit. Am I missing something?

---

## Round 0.2 · accepted · Accept

The revision of the manuscript is acceptable.

Reviewer 1 ·

Basic reporting

The revised manuscript is clearly written in professional logic and structure, the English text is well-improved compared to the last version. Moreover, the revised figure and tables are professional. Apparently, the authors have checked the manuscript thoroughly, and revised every comment carefully.

Experimental design

The model (or method) used in this work is overall adequate, as the other reviewer has pointed out. I think the research design is reasonable and fit for this case study.

Validity of the findings

I noticed that the authors have added several new findings in the revised manuscript. In my point of view, these new findings are closely related to the topic.

Additional comments

The authors have done much work on disposal of or reuse of agricultural straw, it has environmental and economic benefits. And also a bio-economic model –“straw resource utilizaiton for fungi in China” was introduced, which is described with sufficient detail and information. So this work is interesting and is of important significance.

Reviewer 2 ·

Basic reporting

Good.

Experimental design

Good.

Validity of the findings

Good.